# The Association between COVID-19-Related Wellbeing with Materialism and Perceived Threat

**DOI:** 10.3390/ijerph19020912

**Published:** 2022-01-14

**Authors:** Fei Teng, Jiaxin Shi, Xijing Wang, Zhansheng Chen

**Affiliations:** 1Faculty of Psychology, South China Normal University, Guangzhou 510631, China; tengfei.scnu@gmail.com; 2Department of Psychology, The University of Hong Kong, Hong Kong, China; billshi@connect.hku.hk (J.S.); chenz@hku.hk (Z.C.); 3Department of Social and Behavioural Science, City University of Hong Kong, Hong Kong, China

**Keywords:** materialism, perceived threat, wellbeing, anxiety, depression, COVID-19

## Abstract

The ongoing coronavirus disease (COVID-19) pandemic has had a profound impact on people’s wellbeing. Here, we proposed that an individual characteristic might be associated with wellbeing; that is, materialism. Specifically, we conducted three studies (total *N* = 3219) to examine whether people with high levels of materialism would experience poorer wellbeing (i.e., anxiety and depression, in the current case). The results showed that materialism was positively associated with depression (Studies 1A, 1B and 2) and anxiety (Study 2). Moreover, such a relationship was mediated by people’s perceived threat of COVID-19 (Study 2). These findings were observed in both Chinese and American people. The findings are discussed in terms of their theoretical and practical contributions.

## 1. Introduction

We live in a world full of threats, such as unemployment, pandemics, and accidents, which threaten people’s economic wellbeing, health, and even their safety. Regardless of the type of threat, these are perceived as dangerous situations that are harmful to one’s wellbeing (APA Dictionary of Psychology). This raises the important question of who is more vulnerable to these threats and experiences poor wellbeing as a result. Previous studies have suggested that several individual characteristics are negatively associated with aspects of wellbeing, such as emotional stability [1]. In this study, we propose that individuals who have materialistic values and goals would perceive threats more strongly, than experiencing poor wellbeing, such as experiencing anxiety and depression in the current situation.

### 1.1. Antecedents of Wellbeing during the COVID-19 Pandemic

It is not surprising that people’s well-being, in general, is lowered by the pandemic (see [2] for a review), given that various aspects of their life have been negatively impacted. Studies have already revealed that the threat of the Coronavirus disease [3,4], mandatory quarantine [5], and economic uncertainty [6] are robust predictors for reduced well-being. In addition, several studies have identified that the well-being of vulnerable groups could be more subject to the influence. For example, the elderly [7], the homeless [8], and pregnant women [9] have reported experiencing significant mental health issues such as anxiety and depression. In addition to those particular groups, researchers have highlighted several individual and social antecedents that could contribute to negative well-being among general populations. These factors include hierarchical thinking (i.e., social dominance orientation [10]), aggressive online behavior towards stigmatized groups [11], and frequent social media exposure [12]. In the current research, we would like to explore another individual factor that could potentially affect well-being during this pandemic, i.e., materialism., a point to be elaborated on in the following section.

### 1.2. Materialism and Wellbeing

Materialism is defined as a set of values/goals that concern one’s own material possessions [13]. Materialistic individuals tend to equate acquisitions and possessions with happiness, success, and the centrality of life [14]. In other words, materialism inevitably diverts the time, attention, and energy that could otherwise be devoted to other aspects of life, such as family, community, or spiritual growth. This is because an individual’s value/goal system is organized in a circular pattern, with each value consistent with some values but in conflict with others [15]. Specifically, materialism is classified under the extrinsic orientation and self-enhancement in circumplex models [16], which focus on self-oriented and extrinsic values (e.g., wealth success, status, and popularity). On the other end of the human goal system resides the other-oriented and intrinsic values (e.g., community, affiliation, and self-acceptance). According to the self-determination theory [15], if individuals overemphasize the importance of materialism, their other-oriented and intrinsic goals are either neglected or compelled to battle with their extrinsic pursuits.

Indeed, empirical evidence has accumulated to support that materialism impairs other-oriented behavior and interpersonal relationships. For instance, materialistic college students are found to exhibit a lower empathy level, such that these students saw little need to consider others’ minds and perspectives [17]. Moreover, materialistic individuals are less likely to trust others [18], engage less in cooperative and prosocial behaviors [19,20] compared to their non-materialistic counterparts. A longitudinal study found a direct relationship between materialism and loneliness [21]. Furthermore, those who endorse materialism show a higher level of Machiavellianism, a dark personality trait of manipulating and exploiting others to maximize one’s self-interests [22]. More strikingly, even temporarily heightening one’s aspiration for money can lead people to treat others as tools to facilitate personal goal achievement and neglect irrelevant others’ minds, a phenomenon known as objectification [23]. Even intimate relationships are not exempt from its influence. Materialistic individuals report less satisfying family life and friendships [24]. This feeling is reciprocal, such that the relationships are also rated by their friends and family as low quality [25]. Marital quality is the worst among couples who both endorse materialism, relative to those with no or only one materialistic partner [26]. One of the reasons could be that materialism is negatively correlated with the partner’s disclosure and responsiveness levels that are crucial for forming intimacy and bonding [27].

In addition, due to shifted time, attention, and energy to extrinsic values/goals from intrinsic ones, materialism may result in low levels of subjective wellbeing [13,28]. Numerous studies using the values-conflict framework have shown that materialism is negatively linked to various aspects of one’s wellbeing, including fewer positive affect experiences [29], lower life satisfaction [30], and poorer physical health [31]. More relevant to the current work, several studies have revealed a positive relationship between materialism and depression [32,33]. In addition, Kim, Kasser, and Lee, [34] found a positive association between extrinsic values and anxiety in both South Korean participants and U.S. participants. A recent meta-analytic review indicates a modest correlation between materialism and the mental health issues of anxiety and depression [35]. Besides this, the meta-analysis supported a need-based explanation for the negative association; that is, materialism cannot satisfy the psychological needs (e.g., autonomy) that can benefit wellbeing. Materialism is, indeed, detrimental to people’s wellbeing.

Despite that the link between materialism and well-being has been tested multiple times in previous research, no studies focused on any special periods of time (e.g., crisis). Therefore, the first aim of the study is to test whether the effect of materialism on well-being found in previous research could be observed during this pandemic. Of equal importance, we aim to test a (potential) mediator (i.e., perceived threat) in this process, given that much fewer studies have focused on underlying mechanisms (except [35]). We believe that materialism could predict a high level of perceived threat during the pandemic, which could further predict reduced well-being, points to be elaborated on in the following sections.

### 1.3. Materialism, Perceived Threat and Wellbeing

Based on the characteristics of materialism, we argue that people who endorse such a belief will perceive a greater threat when facing a dangerous situation (e.g., pandemic), leading to poor wellbeing. As noted earlier, materialism values extrinsic goals (e.g., financial success); however, people who value material possessions also endure the greatest threats. For example, materialism values wealth success; however, this pursuit becomes very difficult when confronted with a global recession and high unemployment. More importantly, compared to those who are less materialistic, people with higher levels of materialism often encounter interpersonal problems, a point that has been explained in a previous section. Impaired interpersonal relationships can leave them short of the necessary interpersonal resources to cope with the threat Lacking social connectedness can also lead to other undesirable outcomes, such as less cooperation [36]. Obviously, impaired social connectedness results in lessened social support to defend oneself against external threats; thus, such people perceive threats more substantially. In contrast, close relationships work as a buffering mechanism against threats [37]. A previous study has also revealed that perceived social support is negatively associated with the perceived threat of disease [38]. Indeed, preliminary findings have suggested the link between materialism and threat. For instance, research also demonstrated that materialistic preferences appear to be positively associated with existential insecurity in the form of death anxiety. Specifically, Kasser and Kasser [39] analyzed dreams. They found that people high in materialistic values were more likely to report dreams related to death and mortality than people low in materialistic ones. Taken together, we predict that people with high materialism could be more likely to perceive a greater threat due to the conflict between their goals and impaired interpersonal relationships.

The relationship between perceived threat and wellbeing is clear. As experiencing a threat induces a sense of insecurity, it undermines an individuals’ wellbeing. Many studies have supported this idea. For example, perceiving threats is positively associated with worry [40], but negatively associated with self-esteem [41]. People with materialistic goals and values are highly vulnerable to threats, yet they often lack sufficient social support to avoid these threats, which leads them to have poorer wellbeing. The current epidemic caused by the novel coronavirus 2019 (COVID-19) poses a threat to our predictions. This outbreak has spread around the world for more than eight months, causing more than 800,000 deaths and more than 25 million cases [42]. In addition to threatening people’s health, the negative impacts induced by COVID-19 are multifaceted and profound, including economic recession and an increased risk of unemployment. We argue that people who prioritize materialistic goals and values perceive a greater threat in COVID-19 and experience poor wellbeing as a result.

## 2. The Present Research

We focused on the relationship between materialism and wellbeing during the outbreak of COVID-19 across three studies that examined two hypotheses. First, consistent with previous studies (e.g., [43]), we expected that people scoring high in materialism would experience a low level of wellbeing during the current COVID-19 outbreak. Second, the negative association between materialism and wellbeing would be mediated by the increased threat of COVID-19.

### 2.1. Study 1A

In Study 1A, we aimed to test our predicted association between materialism and wellbeing. Specifically, we measured wellbeing using depressive symptoms as an indicator, which has been widely accepted in previous studies (e.g., [44]).

#### 2.1.1. Method

Participants. We recruited 509 participants for this study through a Chinese survey website (www.wjx.com, accessed on 15 August 2020), a participant recruitment platform that is comparable to Amazon Mturk in China. The average age of the participants was 26.18 (*SD* = 9.29) years (Three participants did not indicate their ages.) and 78.2% (*N* = 398) were female. All participants gave their informed consent and received a small payment for their participation. The sensitivity analysis (α = 0.05 and power = 0.80) suggested that the minimum effect size should be 0.02 based on the current sample size.

Procedures and measures. We required participants to fill out several questionnaires to measure their materialism and negative affect. Unless noted otherwise, the items were rated on a seven-point Likert-type scale. After that, participants provided their basic demographic information (i.e., sex, age, and subjective status). Finally, participants were thanked and debriefed.

Materialism. We measured materialism using the nine-item Material Value Scale (MVS; [45]), which included items such as “*I admire people who own expensive homes, cars, and clothes*”, and “*I like to own things that impress people*”. Responses were taken using a seven-point scale ranging from 1 (*extremely uncharacteristic of me*) to 7 (*extremely characteristic of me*). All items were averaged to form an index, with higher scores indicating higher levels of materialism (α = 0.794).

Depressive symptoms. We measured depressive symptoms using a three-item scale adapted from [46]. The items were “*Recently, I usually feel depressed*”, “*Recently, I feel upset easily*”, and “*Recently, I can easily deal with pressure*”. All responses were taken using a seven-point scale, ranging from 1 (*strongly disagree*) to 7 (*strongly agree*). All scores were averaged to form an index, and a higher score indicated a higher level of depressive symptoms (α = 0.750).

Subjective social status. Participants indicated their subjective social status using the MacArthur Scale [47]. There was a ladder with 10 rungs representing 10 social statuses of China’s society from the bottom to the top. Participants were required to indicate the rung in which they perceived themselves to be located.

#### 2.1.2. Results and Discussion

We presented descriptive statistics and correlations of each study variable in Table 1. The results showed that materialism was positively associated with depressive symptoms, r (509) = 0.30, *p* < 0.001, 95% CI [0.22, 0.38].

After controlling for participants’ demographic variables, including sex, age, and subjective social status, materialism still significantly predicted depressive symptoms (see Table 2).

The current findings supported our hypothesis that materialism was positively related to depressive symptoms.

### 2.2. Study 1B

In Study 1B, we aimed to replicate the findings of Study 1A using participants from the United States.

#### 2.2.1. Method

**Participants.** We recruited 701 participants through Amazon Mturk (365 males; 52.1%). The average age was 42.53 (*SD* = 12.83) years. Among the participants, 73.7% were White, 15.5% were African American, 6.7% were Asian, and 4.1% were of other races. All participants gave their informed consent and received monetary compensation for their participation. The sensitivity analysis (α = 0.05 and power = 0.80) suggested that the minimum effect size should be 0.02 based on the current sample size.

**Procedures and measures**. As in Study 1A, we measured the participants’ materialism, depressive symptoms, and took their basic demographic information (i.e., sex, age, and subjective social status). Finally, participants were thanked.

**Materialism.** Materialism was measured the same way as in Study 1A (*α* = 0.911).

**Depressive symptoms.** We measured participants’ depressive symptoms using a four-item scale adapted from [46]. Example items were “*Recently, I have frequent mood swings*”, “*Recently, I usually feel depressed*”, “*Recently, I can easily deal with pressure (reversed)*”, *and* “*Recently, I get upset easily*”. Participants’ responses were given on a seven-point scale ranging from 1 (*strongly disagree*) to 7 (*strongly agree*). We averaged all items to indicate negative affect, with a higher score indicating a higher level of negative affect (α = 0.754).

**Subjective social status.** We measured subjective social status using the MacArthur Scale, identical to Study 1A.

#### 2.2.2. Results and Discussion

We presented descriptive statistics and correlations of each variable in Table 3. The results again showed that materialism was positively linked to depressive symptoms, *r* (701) = 0.37, *p* < 0.001, 95% CI [0.30, 0.43].

The results remained the same after controlling for participants’ demographic variables, including sex, age, and subjective social status (see Table 4).

Combining Studies 1A and 1B, we found that materialism was positively associated with depressive symptoms. In the following study, we further identified the role of the perceived threat of COVID-19 (PTC) as a potential mediator of the main effect.

### 2.3. Study 2

Study 2 improved Studies 1A and 1B in two ways. First, we used two validated scales to measure participants’ mental health (i.e., anxiety and depression). Second, we measured participants’ PTC to examine its role in mediating the relationship between materialism and anxiety and depression.

#### 2.3.1. Method

**Participants**. We recruited 2008 participants for this study through a Chinese survey website (www.wjx.com, accessed on 15 August 2020), a participant recruitment platform that is comparable to Amazon Mturk in China. The average age of the participants was 21.49 (SD = 5.23) years (One hundred and eighty-nine participants did not indicate their ages), and 69.7% (*N* = 1400) were female. Participants gave their informed consent and received payment for their participation. Sensitivity analysis (α = 0.05 and power = 0.80) suggested that the minimum effect size should be 0.005 based on the current sample size.

**Procedures and measures**. Similar to Study 1, participants were instructed to fill out a few questionnaires regarding their materialism, anxiety, and depression, as well as other necessary demographic information (i.e., sex, age, and subjective status). We also measured their PTC. Finally, they were thanked and fully debriefed.

**Materialism**. We measured participants’ materialism using a four-item scale taken from the MVS [45]. An example item was “*I like a lot of luxury in my life*”. All responses were given on a seven-point Likert scale, ranging from 1 (*extremely uncharacteristic of me*) to 7 (*extremely characteristic of me*). We averaged the scores to form an index (α = 0.687), with higher scores corresponding to higher levels of materialism.

**Anxiety and depression**. Anxiety and depression were measured using two scales, the Hospital Anxiety and Depression (HAD) scale [48] and the Patient Health Questionnaire (PHQ; [49]). The HAD, a 14-item scale, includes two sub-scales to measure participants’ anxiety and depression, respectively, and the PHQ, a nine-item scale, is also used to measure depression. Prior studies have shown that these measures can be administered to the general population [50,51]. Due to the current COVID-19 situation, we slightly modified the items. The sample items of the two scales were “*During the outbreak of COVID-19, I feel down, depressed, or hopeless*” and “*During the outbreak of COVID-19, I feel tense or wound up*”, respectively. The Cronbach’s α of the scales were 0.901 (PHQ), 0.751 (HAD-A), and 0.722 (HAD-D). Participants responded on a four-point Likert scale ranging from 0 (*not at all*) to 3 (*very often*). The final scores were averaged to indicate participants’ anxiety and depression levels.

**The perceived threat of COVID-19**. We measured participants’ PTC by asking four questions: “How likely are you to be infected by COVID-19?”, “How much fear do you feel about COVID-19?”, “How contagious do you think COVID-19 is?”, and “How much do you think COVID-19 affects mortality rate?” Participants responded on a seven-point Likert scale ranging from 1 (not at all) to 7 (very much; α = 0.633).

**Subjective social status**. Similar to Study 1A, participants indicated their subjective status (*M* = 4.61, *SD* = 1.64).

#### 2.3.2. Results and Discussion

Descriptive statistics and correlations are presented in Table 5. It was found that materialism was positively associated with anxiety and depression. Specifically, materialism was positively associated with HAD-A, *r*(2008) = 0.08, *p* < 0.001, 95% CI [0.04, 0.12], HAD-D, *r*(2008) = 0.1, *p* < 0.001, 95% CI [0.06, 0.14], and PHQ, *r*(2008) = 0.22, *p* < 0.001, 95% CI [0.18, 0.26].

Next, we performed a regression analysis on the effects of materialism on anxiety and depression, controlling for participants’ demographic variables, including sex and subjective social status. We did not include age in the regressions as 189 participants did not indicate their ages, and age was not associated with differences in PTC, anxiety, or depression. The results showed that materialism positively predicted anxiety and depression (see Table 6, Table 7 and Table 8).

#### 2.3.3. Mediation Tests

We examined the extent to which anxiety (measured using the HAD-A) could be explained by participants’ PTC. Materialism predicted PTC (B = 0.11, SE = 0.03, *p* < 0.001) and PTC predicted anxiety (B = 0.60, SE = 0.05, *p* < 0.001). The bootstrap analysis (5000 times) of the indirect effect indicated that PTC was a significant mediator (IE = 0.07, 95% CI [0.03, 0.08], excluding *zero*). Please see Figure 1A for details.

We also examined the extent to which depression (measured by the HAD-D) could be explained by PTC. Materialism predicted PTC (B = 0.11, SE = 0.03, *p* < 0.001) and PTC predicted depression (B = 0.33, SE = 0.05, *p* < 0.001). The bootstrap analysis (5000 times) of the indirect effect indicated that PTC was a significant mediator (IE = 0.04, 95% CI [0.02, 0.06], excluding *zero*). Please see Figure 1B for details.

Lastly, we examined the extent to which depression (measured using PHQ) could be explained by PTC. Following the same steps as above, the results showed that materialism predicted PTC (B = 0.11, SE = 0.03, *p* < 0.001), and that PTC predicted depression (B = 0.50, SE = 0.08, *p* < 0.001). The bootstrap analysis (5000 times) of the mediation effect indicated that PTC was a significant mediator (IE = 0.05, 95% CI [0.03, 0.08], excluding *zero*). Please see Figure 1C for details.

In addition, fear alone could act as a mediator to account for the effect of materialism on anxiety and depression (Materialism positively predicted fear (B = 0.24, SE = 0.04, *p* < 0.001) and fear positively predicted anxiety and depression (HAD-A: B = 0.55, SE = 0.03, *p* < 0.001, HAD-D: B = 0.31, SE = 0.04, *p* < 0.001; depression: B = 0.40, SE = 0.05, *p* < 0.001). The bootstrap analysis (5000 times) of the indirect effect indicated that fear was a significant mediator (IE = 0.13, 95% CI [0.08, 0.18]; IE = 0.07, 95% CI [0.04, 0.10]; IE = 0.10, 95% CI [0.05, 0.14], respectively).).

In summary, PTC mediated the positive associations between materialism and anxiety and depression, which supported our hypothesis that participants with higher materialism would perceive a greater threat of COVID-19, then decreasing their level of wellbeing.

## 3. Discussion

Threats influence people’s lives and wellbeing and these can be found everywhere. The COVID-19 pandemic has threatened people’s physical health, wellbeing, and job safety, among other aspects. Naturally, people have developed feelings of anxiety and depression due to the outbreak. In this study, we found that such reactions were more salient for people who possessed highly materialistic values and goals; that is, people with higher levels of materialism reported a higher level of depression and anxiety (Studies 1A, 1B, and 2). We also identified the mediating role of the perceived threat of COVID-19 (PTC) on the relationship between materialism and reduced mental health (Study 2). Moreover, these effects emerged in both Chinese and American participants.

The above results are consistent with the findings obtained in previous studies which showed that materialism is negatively associated with personal wellbeing [52]. Although feelings of anxiety and depression seem natural during an outbreak, our studies suggest that materialism can also contribute to this process by increasing the PTC. In other words, our research suggests that not just belonging to a vulnerable group (e.g., the homeless and the pregnant women, refs. [8,9] makes one more subject to the negative influence of the pandemic, holding certain values or pursuing certain goals, in our case materialistic ones, increases one’s mental health risks. Therefore, our research contributes to a growing body of studies revealing individual and social antecedents to negative wellbeing among general populations (e.g., [10,11,12]).

We have found that materialism predicts a greater level of perceived threat during the pandemic. In other words, our study suggests that in threatening situations, people who emphasize monetary values and goals are more sensitive to these potentially harmful factors. Thus, they may have a strong loss aversion in such an environment. Interestingly, Ardnt, Solomon, Kasser, and Shelton [53] adopted a Terror Management perspective on materialism and proposed that people may generate materialistic values to defend the existential anxiety surrounding the death issue. Correspondingly, research cumulated to demonstrate that people resorted more to material possessions when death anxiety is salient (e.g., [54,55]), probably as a strategy to cope with existential fear (e.g., [56]). However, our research, together with others (see [52] for a review), suggests that materialism may not be a good strategy to cope with anxiety and fear since materialism per se could be linked with increased mental vulnerability during a crisis.

In addition, since threats have been linked to other psychological challenges such as prejudice [57], stigmatization [58], aggression [59], and even dehumanization (i.e., treating human targets as less than humans, [60]), highly materialistic people may have strong inclinations to demonstrate these negative cognitive and behavioral patterns. A previous study has found that inappropriate behavior towards others (e.g., online aggressive behavior) during a pandemic can predict impaired well-being among perpetrators (Teng et al., 2021). In other words, these subsequent cognitive and behavioral outcomes triggered by threats could further lower the well-being of materialistic individuals. Future studies should examine these possibilities.

As COVID-19 continues to spread, it is extremely important to reduce its negative impact on our lives. Our study provides valuable insights for individuals by revealing the association between materialism and a low level of wellbeing, which should prompt people to decrease their materialistic beliefs. Kasser (2016) has shown a few useful strategies to decrease materialistic values and goals. For example, encouraging people to focus more on the values and goals which oppose materialism (e.g., community and interpersonal connection) may be helpful, and this is in line with the recommendations of the WHO [61]. In addition, a recent study found that exposure to nature can reduce materialism [62]. This reminds us that the benefits of connecting with nature may not only function in enhancing wellbeing but also help in reshaping our values and goals by reducing the importance of extrinsic values.

### Limitations and Future Directions

Despite these implications, the study has several limitations. One major limitation is that we conducted only cross-sectional and correlational studies to examine the association between materialism and wellbeing. In line with Kasser [13], future studies can adopt a longitudinal and experimental methodology to better examine the effect of the current situation. Furthermore, in our study, personal wellbeing relies solely on self-reports. Although some studies have indicated similar results between the self-report and interviewers’ perspectives, we still recommend more studies that can use various measures to capture one’s wellbeing (e.g., diary recording). Lastly, we only examined the threat of COVID-19 and did not consider the possibility of other types of threats. Although the impact of COVID-19 is multifaceted, including threats to health and wealth, future studies can further test the association between materialism and perceived threat under different situations.

## 4. Conclusions

The outbreak of COVID-19 has inevitably threatened several aspects of people’s lives. An emerging body of studies has revealed that belonging to a vulnerable group (e.g., the homeless and the pregnant women [8,9]) makes one more subject to the negative influence of the pandemic. In the current research, we tested whether being materialistic could also increase one’s mental health issues. Across three studies, we found that consistent with previous studies (e.g., [13]), people scoring high in materialism experience a low level of wellbeing during the current COVID-19 outbreak. In addition, the negative association between materialism and wellbeing can be mediated by the increased COVID-19 threat perception. Therefore, our research identified an antecedent contributing to negative wellbeing among general populations and suggests the importance of holding and pursuing the appropriate values and goals during a crisis.

## Figures and Tables

**Figure 1 ijerph-19-00912-f001:**
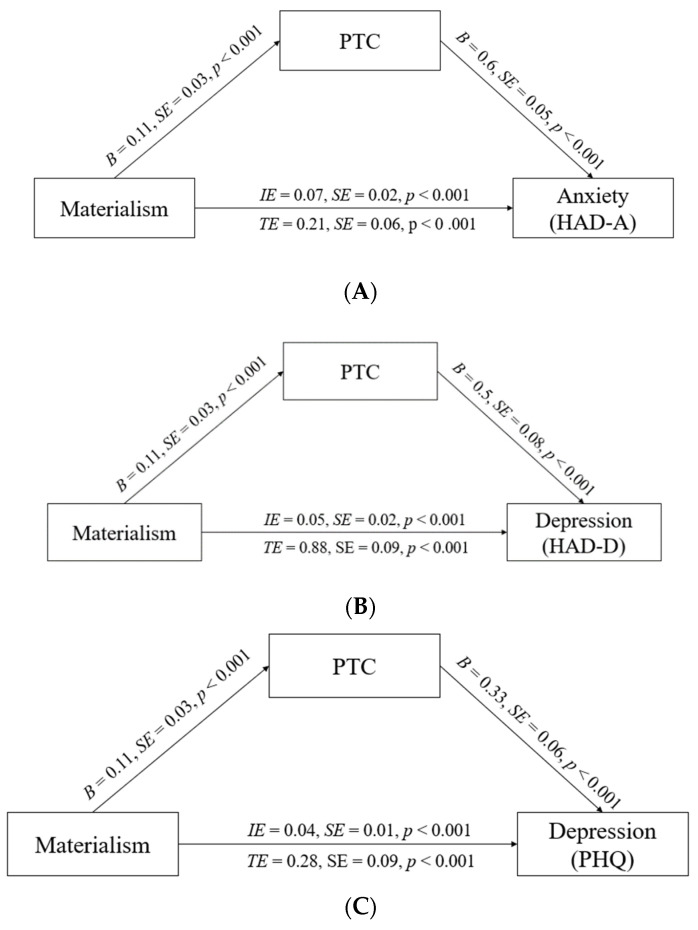
(**A**). Mediation model for the effect of materialism on anxiety (measured by HAD-A) via PTC, Study 2. Un-standardized regression coefficients are shown. (**B**). Mediation model for the effect of materialism on depression (measured by HAD-D) via PTC, Study 2. Un-standardized regression coefficients are shown. (**C**). Mediation model for the effect of materialism on depression (measured by PHQ) via PTC, Study 2. Unstandardized regression coefficients are shown. PTC = Perceived threat to COVID-19, IE = indirect effect, TE = total effect.

**Table 1 ijerph-19-00912-t001:** Descriptive Statistics and Correlations of Each Variable (Study 1A).

Variables	*M (SD)*	1	2	3	4
1. Age	26.18 (9.29)	-			
2. Subjective social status	4.82 (1.56)	0.18 ***	-		
3. Materialism	4.28 (0.99)	−0.15 ***	−0.04	-	
4. Depressive symptoms	3.90 (1.35)	−0.19 ***	−0.17 ***	0.3 ***	-

Note: *** *p* < 0.001.

**Table 2 ijerph-19-00912-t002:** Regression Results Predicting Depressive Symptoms (Study 1A).

Variables	B	SE	β	*t*	*p*
Sex (0 = male; 1 = female)	0.31	0.14	0.09	2.23	0.026
Age	−0.02	0.01	−0.11	−2.53	0.012
Subjective social status	−0.12	0.04	−0.13	−3.19	0.002
Materialism	0.37	0.06	0.27	6.54	<0.001
	*R*^2^ = 0.14				
	*F*(4, 501) = 20.13, *p* < 0.001				

**Table 3 ijerph-19-00912-t003:** Descriptive statistics and correlations of each variable (Study 1B).

Variables	*M (SD)*	1	2	3	4
1. Age	42.58 (12.70)	-			
2. Subjective social status	5.47 (2.01)	−0.11 **	-		
3. Materialism	3.71 (1.46)	−0.32 ***	0.31 ***	-	
4. Depressive symptoms	3.03 (1.43)	−0.26 ***	0.01	0.37 ***	-

Note: ** *p* < 0.01, *** *p* < 0.001.

**Table 4 ijerph-19-00912-t004:** Results of regression analysis of factors predicting depressive symptoms (Study 1B).

Variables	B	SE	β	*t*	*p*
Sex (0 = male; 1 = female)	0.11	0.10	0.04	1.12	0.264
Age	−0.02	0.00	−0.17	−4.57	<0.001
Subjective social status	−0.08	0.03	−0.11	−3.11	0.002
Materialism	0.34	0.04	0.35	9.26	<0.001
	*R*^2^ = 0.17				
	*F*(4, 696) = 35.91, *p* < 0.001				

**Table 5 ijerph-19-00912-t005:** Descriptive statistics and correlations of each variable (Study 2).

Variables	*M (SD)*	1	2	3	4	5	6	7
1. Age	21.49 (5.23)	-						
2. SSS	4.61 (1.4)	0.06 **	-					
3. Materialism	3.95 (1.13)	−0.04	0.04	-				
4. PTC	5.02 (1.31)	0.01	−0.08 ***	0.09 ***	-			
5. Anxiety (HAD-A)	4.4 (2.90)	0.04	−0.1 ***	0.08 ***	0.28 ***	-		
6. Depression (HAD-D)	4.37 (3.14)	0.03	−0.16 ***	0.1 ***	0.15 ***	0.66 ***	-	
7. Depression (PHQ)	5.70 (5.09)	−0.02	−0.11 ***	0.22 ***	0.16 ***	0.53 ***	0.59 ***	-

Note: ** *p* < 0.01, *** *p* < 0.001; SSS = Subjective social status, PTC = Perceived threat of COVID-19, HAD-A = Anxiety subscale of the Hospital Anxiety and Depression scale, HAD-D = Depression subscale of the Hospital Anxiety and Depression scale, PHQ = Patient Health Questionnaire.

**Table 6 ijerph-19-00912-t006:** Results of regression analysis of factors predicting anxiety (HAD-A; Study 2).

Variables	B	SE	β	*t*	*p*
Sex (0 = male; 1 = female)	0.16	0.14	0.03	1.12	0.263
Subjective social status	−0.19	0.04	−0.11	−4.78	<0.001
Materialism	0.22	0.06	0.08	3.85	<0.001
	*R*^2^ = 0.02				
	*F*(3, 2004) = 12.44, *p* < 0.001				

**Table 7 ijerph-19-00912-t007:** Results of regression analysis on factors predicting depression (HAD-D; Study 2).

Variables	B	SE	β	*t*	*p*
Sex (0 = male; 1 = female)	−0.52	0.15	−0.08	−3.50	<0.001
Subjective social status	−0.30	0.04	−0.16	−7.15	<0.001
Materialism	0.29	0.06	0.11	4.78	<0.001
	*R*^2^ = 0.04				
	*F*(3, 2004) = 28.46, *p* < 0.001				

**Table 8 ijerph-19-00912-t008:** Results of regression analysis on factors predicting depression (PHQ; Study 2).

Variables	B	SE	β	*t*	*p*
Sex (0 = male; 1 = female)	−0.10	0.22	−0.01	−0.45	0.652
Subjective social status	−0.34	0.06	−0.12	−5.63	<0.001
Materialism	0.90	0.09	0.22	10.28	<0.001
	*R*^2^ = 0.06				
	*F*(3, 2004) = 44.57, *p* < 0.001				

## Data Availability

The data presented in this study is fully available on request from the corresponding author.

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
