# Peer review of "The Association between COVID-19-Related Wellbeing with Materialism and Perceived Threat"

_ijerph, 2022, doi:10.3390/ijerph19020912_

Round 1

Reviewer 1 Report

Good work, thanks for the work. The new contributions and corrections are more appropriate and provide a greater understanding of the study. Thanks

Reviewer 2 Report

The authors have made numerous changes, additions, to the article, which have improved the article. It is now much better, more scientific, coherent and complete.

They have 'corrected' or remedied the shortcomings that I considered, especially in the section on the method (completed with tables) and conclusions (conveniently modified section). The authors have also revised other aspects indicated in the initial evaluation (limitations, discussion...).

With the changes made, I consider the article publishable.

This manuscript is a resubmission of an earlier submission. The following is a list of the peer review reports and author responses from that submission.

Round 1

Reviewer 1 Report

This analysis could be much more specific, but very briefly it could be summed up as follows:

  • Topic and problem not interesting. Although everything related to COVID-19 is highly topical, in this case I think it does not to be of the greatest interest.
  • Little substantiation of the state of the question. Studies and research on the subject are scarce (there are only 34 bibliographical references).
  • Methodologically improvable, it should be better explained (reliability of the sample...).
  • Poor conclusions.

Reviewer 2 Report

The conclusions do not sufficiently develop the object of the study nor are they strong enough. Reconsider completing this section. Thanks

Reviewer 3 Report

The authors present the findings of several studies linking materialism and perceived threat to different aspects of mental health in Chinese and US-American online samples. The manuscript is clearly written, the methodology is sound, and the conclusions are justified by the data.

In revising the manuscript, I would encourage the authors to address the following comments:

Threat and materialism have previously been linked in research on Terror Management Theory, e.g., Arndt, Solomon, Kasser, and Sheldon (2004), published in JCM. These and related findings from that research area may provide additional insights into the psychological functioning of materialism as a buffer against death anxiety, which may lend further credence to the authors claim. In any case, given the conceptual overlaps, it appears appropriate to thoroughly discuss literature from this research area.

Furthermore, some specification of the statistical packages used for conducting the mediation analysis would be helpful for the interested reader. On a related note, I also wonder whether the inclusion of a threat mean score (i.e., PTC) in the analysis is justified given the low internal consistency of the scale (i.e., .63). Given the sample size, the low internal consistency combined with the rather small indirect effect estimates could suggest that the different items of the scale (i.e., threat of infection, fear, assessment of contagiousness, impact on mortality) contribute differently to the mediation pattern. It would be helpful if the authors provide complementary analyses that explore mediation patterns on an item-level. Based on the authors hypothesis, I would expect the fear-item to emerge as (the most) important mediator.